# Loss of SDHB Induces a Metabolic Switch in the hPheo1 Cell Line toward Enhanced OXPHOS

**DOI:** 10.3390/ijms23010560

**Published:** 2022-01-05

**Authors:** Mouna Tabebi, Ravi Kumar Dutta, Camilla Skoglund, Peter Söderkvist, Oliver Gimm

**Affiliations:** 1Department of Biomedical and Clinical Sciences (BKV), Linköping University, 581 83 Linköping, Sweden; ravi.kumar.dutta@liu.se (R.K.D.); camilla.skoglund@liu.se (C.S.); peter.soderkvist@liu.se (P.S.); 2Clinical Genomics Linköping, Science for Life Laboratory, Linköping University, 581 83 Linköping, Sweden; 3Department of Surgery and Department of Biomedical and Clinical Sciences (BKV), Linköping University, 581 83 Linköping, Sweden; oliver.gimm@liu.se

**Keywords:** SDHB, PCCs/PGLs, hPheo1, OXPHOS, glutamine, GLUD1

## Abstract

Background: Enzymes of tricarboxylic acid (TCA) have recently been recognized as tumor suppressors. Mutations in the SDHB subunit of succinate dehydrogenase (SDH) cause pheochromocytomas and paragangliomas (PCCs/PGLs) and predispose patients to malignant disease with poor prognosis. Methods: Using the human pheochromocytoma cell line (hPheo1), we knocked down SDHB gene expression using CRISPR-cas9 technology. Results: Microarray gene expression analysis showed that >500 differentially expressed gene targets, about 54%, were upregulated in response to SDHB knock down. Notably, genes involved in glycolysis, hypoxia, cell proliferation, and cell differentiation were up regulated, whereas genes involved in oxidative phosphorylation (OXPHOS) were downregulated. In vitro studies show that hPheo1 proliferation is not affected negatively and the cells that survive by shifting their metabolism to the use of glutamine as an alternative energy source and promote OXPHOS activity. Knock down of SDHB expression results in a significant increase in *GLUD1* expression in hPheo1 cells cultured as monolayer or as 3D culture. Analysis of TCGA data confirms the enhancement of *GLUD1* in *SDHB* mutated/low expressed PCCs/PGLs. Conclusions: Our data suggest that the downregulation of SDHB in PCCs/PGLs results in increased *GLUD1* expression and may represent a potential biomarker and therapeutic target in SDHB mutated tumors and SDHB loss of activity-dependent diseases.

## 1. Introduction

Succinate dehydrogenase (SDH) is a mitochondrial enzyme complex composed of four subunits: SDHA, SDHB, SDHC, and SDHD, as well as the required assembly factor (SDHAF2). SDH is involved in the mitochondrial tricarboxylic acid (TCA) cycle, in the electron transport chain enzyme complex, and the mitochondrial complex II function. Loss of SDH enzyme activity and expression, due to inactivating mutations in the SDH genes, causes pheochromocytoma and paraganglioma and contributes to other tumors, such as renal cell and colorectal carcinomas acting as tumor suppressor genes [1,2,3,4,5]. 

Pheochromocytoma (PCC) and paraganglioma (PGL) are rare neuroendocrine tumors (incidence about 0.8/100,000 person-years) that arise in the adrenal medulla and in paraganglia of the autonomous nervous system, respectively [6]. SDHB mutations, the most frequently inactivated gene among the four subunits of SDH, are associated with malignancy in 17–40% of patients and poor prognosis [6,7], but the mechanisms leading to the malignant phenotype are still unclear.

Inactivating mutations abolish SDH activity, resulting in accumulation of succinate, an oncometabolite, that elicits a pseudohypoxic phenotype and contributes to cancer development [8,9].

Indeed, succinate competitively inhibits the 2-oxoglutarate (2-OG)-dependent HIF prolyl-hydroxylases in the cytosol, which results in the stabilization and activation of the hypoxia-inducible transcription factors, HIF1 and 2 [6,10]. In SDHx mutated tumors, metabolic abnormalities such as increased lactate dehydrogenase A (LDHA) expression and lactate production have been observed [11,12]. Of interest, SDHB mutated tumors showed dysregulation of genes involved in proliferation, adhesion, and hypoxic pathway [13].

In tumor cells, the mitochondrial ATP decreased; however, the demand for biosynthetic precursors increased, which make elevated glutaminolysis a pathway for certain cancers to compensate for these changes and to maintain functional TCA and oxidative phosphorylation (OXPHOS) [6,13].

Glutamine is a major source of carbon for nucleotide and nonessential amino acid biosynthesis, and its metabolism is involved in cell proliferation [14]. It is taken up by the cells through transporters (SLC1A5/SLC7A5), then converted to glutamate and further to alpha-ketoglutarate (α-KG) by glutaminase 1 or 2 (GLS1/2), glutamate dehydrogenase (GLUD1), glutamate oxaloacetate transaminases (GOT), and other enzymes to resume ATP production through the TCA cycle [13,15].

It was recently shown that the importance of the glutaminolysis pathway promotes proliferation of SDHB-mutated chromaffin cells (PC12, rat pheochromocytoma cell line), suggesting that SDH-associated malignant potential of PCC/PGL is dependent on glutaminase expression [6].

The human pheochromocytoma cell line hPheo1 is the first and only human hTERT-immortalized pheochromocytoma cell line with a normal diploid chromosomal set except for a small deletion of the 9p region, encompassing the *CDKN2A* gene encoding the p16 and p14^ARF^ proteins. The hPheo1 cell line synthesizes and secretes neuroendocrine biomarkers (chromogranin A and Phenylethanolamine N-methyltransferase “PNMT”), previously described by Ghayee and colleagues [16]. Exome sequencing and microarray gene expression analysis showed no mutations and exhibited a normal copy number in the SDHB gene.

In this study, we aimed to investigate, for the first time, the biological effect of knocking down the SDHB gene in the hPheo1 cell line using CRISPR-cas9 technology. Our study revealed that knockdown (KD) of SDHB has a positive effect on cell proliferation. In addition, we show that KD-SDHB in human chromaffin cells adapt to more rapid metabolism, which allows these cells to survive by shifting their metabolism to the use of an alternative fuel, glutamine.

## 2. Results

### 2.1. Downregulating SDHB in hPheo1 Cell

The establishment of the KD-SDHB hPheo1 cell line using CRISPR-cas9 technology led to the generation of a heterozygous deletion of 20 nucleotides in exon 2 (c.142_161del; *p. Asp48Glnfs*7*) resulting in a frame shift mutation giving a short protein of 54 aa instead of 280 aa (Appendix A). To confirm the knockdown on both the RNA and the protein level, ddPCR and western blot was performed showing a 75% reduction in mRNA levels and SDHB protein expression (Figure 1A,B). Attempts to generate deleted homozygous *SDHB* (knock-out *SDHB*) cells failed, and the cells die quickly after transfection.

By measuring extracellular metabolite levels, we noticed a significant increase in lactate with a four-fold increase in KD-SDHB compared to hPheo1 control cells (*p* = 0.0012) (Figure 1C). The increased lactate level reflects enhanced glycolytic flux.

To understand the consequences of SDHB deficiency on gene expression, a whole genome expression analysis was performed. A fold change of 2, *p* < 0.05, and FDR *p* < 0.2 were considered to determine the number of genes to examine gross changes in expression. This yielded a final sample set of 520 targets, about 54% of which were upregulated in response to SDHB deficiency. Analysis of gene expression levels showed that these genes were differentially regulated in both control and deficient cells. Using gene set enrichment analysis, we were able to assign these genes to biological functions. Notably, genes involved in glycolysis, hypoxia, cell proliferation, and cell differentiation (i.e., PI3AKT, MYC targets, Collagen formation, and Extra cellular matrix (ECM)) were up regulated in this analysis, whereas genes involved in oxidative phosphorylation (OXPHOS) were downregulated (Figure 1D,E). 

A substantial number of the upregulated genes in the enriched Hallmark gene sets were involved in hypoxia, epithelial-mesenchymal transition (EMT), glycolysis, and mTORC signaling, whereas several of the downregulated genes were related to oxidative phosphorylation (OXPHOS) (Figure 1D,E). Furthermore, from the C2: Canonical pathways collection and among many of the upregulated genes, we found three highly enriched gene sets which included hypoxia, glycolysis, and glucose metabolism, as well as extracellular matrix (ECM) components and related proteins, or the so-called “matrisome pathway.” These results suggest that the SDHB-deficient cells exhibit reduced aerobic metabolism and enhanced glycolysis. 

To assess KD-SDHB in a more relevant in vitro model, we applied 3D culturing of hPheo1 cells by spheroid formation (Appendix A). Microarray analysis and GSEA showed very similar gene expression patterns and involved pathways as the 2D cell culture (Appendix A).

### 2.2. Effects in Cell Proliferation 

The microarray data showed that cyclin D1(CCND1) is differentially increased in KD-SDHB hPheo1 cells. Validation by RT-qPCR was performed, and the cell proliferation rate was slightly but not significantly increased (*p* = 0.219 > 0.05) (Figure 2A). On the other hand, the proliferation assay showed a significantly higher proliferation rate (*p* < 0.05) in KD-SDHB cells (Figure 2B). Clonogenic assay showed that the KD-SDHB forms more clones than the hPheo1 cells, suggesting a proliferating ability and active metabolism for those cells (Appendix A).

### 2.3. Effects on Cellular Adhesion

Transcriptomic data showed that SDHB function in hPheo1 cells affects ECM and adhesion’s function. Validation with RT-qPCR was performed for several genes, and we detected ≥ 2-fold upregulation of the transcription factor Glu/Asp-rich carboxy-terminal domain 2 (*CITED2*), transforming growth factor-β3 (*TGFB3*), Collagen Type I Alpha 2 Chain (*COL1A2*), and Collagen Type VI Alpha 3 Chain (*COL6A3*) (Figure 3A). Besides, a gene set assembly of collagen fibrils and other multimeric structures was enriched, and genes involved in collagen remodeling, such as LOXL, were upregulated in the KD-SDHB cells (Appendix A). 

Furthermore, we performed a trypsin sensitivity test, which represents impaired expression on extracellular proteins of trypsin binding sites, affecting the ECM affinity through the alteration of adhesion complex expression and cellular receptor conformational changes. We found that *KD-SDHB* resulted in a 1.7-fold increase in the percentage of trypsin-sensitive cells and a 2.3-fold decrease in the percentage of trypsin-resistant cells (Figure 3B). 

### 2.4. Effects in Mitochondrial Respiration

Since transcriptomic data and GSEA showed downregulation in gene sets associated with oxidative phosphorylation and mitochondrial respiratory electron transport, we performed further experiments to determine those effects. The mtDNA copy number showed a five-fold decrease in KD-SDHB cells compared to hPheo1 cells (Figure 4A), which correlates with the decreased expression levels of mitochondrial genes observed from the microarray data. On the other hand, peroxisome proliferator-activated receptor gamma coactivator 1-alpha (*PCG1-α*), master regulator of the mitogenesis and involved in the regulation of the cellular energy metabolism, showed ≥2-fold upregulation in the KD-SDHB cells (Figure 4B). It has been demonstrated that a reduction in the number of mitochondria leads to tumorigenesis by inducing changes in redox status, membrane potential, ATP levels, and gene expression [17]. We next assessed the mitochondrial membrane electrical gradient (ψm) by using the dye TMRE to monitor Δψm and observed a significant decrease (*p* < 0.001) in the KD-SDHB cells compared to control hPheo1 cells (Figure 4C), suggesting impaired OXPHOS activity in the cells. 

To assess the OCR which mainly reflects the mitochondrial oxidative phosphorylation (OXPHOS) activity, we used the Seahorse XF24 analyzer. KD-SDHB cells showed a significantly higher basal OCRs compared to control hPheo1 cells (*p* = 0.0005). Furthermore, maximal OXPHOS capacity (*p* < 0.0001) and ATP production (*p* = 0.005) significantly increased in KD-SDHB cells, indicating increased mitochondrial OXPHOS activity. Besides, the non-mitochondrial respiration increased in KD-SDHB cells (*p* < 0.0001) compared to control cells (Figure 4D,E). Elevated mitochondrial respiration was not underpinned by the downregulation of mitochondrial genes involved in OXPHOS (OCR mean = 391 ± 35 vs. 231 ± 13 in hPheo1 cells). 

On the other hand, OCR of SDHB-KD cells was more severely affected by Rotenone/Antimycin (Complex I/III inhibitors) compared to control cells (Figure 4D), which indicates that OXPHOS is more activated. In addition, the non-mitochondrial respiration fraction has a significant effect related to OCR, suggesting that these cells also use non-mitochondrial respiration for survival.

### 2.5. Effects in Glycolysis

A set of genes associated with glycolysis was investigated with RT-qPCR to validate the upregulation of the glycolysis pathway (Figure 5A). From the microarray gene expression profiles, we identified a more than 3-fold upregulation of Glucose transporter 1 (*GLUT1)*, Aldolase-Fructose-Bisphosphate A *(ALDOA)*, Aldolase-Fructose-Bisphosphate C *(ALDOC)*, Pyruvate kinase M2 (*PKM2)*, and Lactate Dehydrogenase A (*LDHA*) (Figure 5B). The enhanced expression of genes involved in glycolysis pathway was accompanied with a decrease in glycolytic activity (ECAR) in KD-SDHB cells (ECAR mean = 39 ± 2 vs. 44.6 ± 2.1 in hPheo1 cells). 

We noted that KD-SDHB cells exhibited a higher growth rate in normal glucose condition compared to control hPheo1 cells. In addition, KD-SDHB cells were less susceptible to 2-deoxy-glucose (2DG), a glucose analog that inhibits glycolysis by hexokinase 2 inhibition (Figure 5C), which suggest that glycolysis is less activated. Indeed, glycolysis was decreased significantly (*p* < 0.0001) in KD-SDHB, but the non-glycolytic acidification increased significantly (*p* = 0.0012) compared to the hPheo1cells, which suggests the presence of another source of extracellular acidification than glycolysis, such as glutaminolysis (Figure 5C,D).

### 2.6. Effects in Glutaminolysis 

The mRNA levels of selected genes involved in the glutamine metabolic pathway were assessed by RT-qPCR (Figure 6A). We observed a four-fold upregulation of Glutamate Dehydrogenase 1/Glutamine synthetase (GLUD1) and a two-fold upregulation of Glutamate-Ammonia Ligase (GLUL) and Glutamic-Oxaloacetic Transaminase 2 (GOT2). However, Solute Carrier Family 1 Member 5 (SLC1A5) and Glutaminase 1 (GLS1) did not change significantly (Figure 6B).

Similar results were obtained, with about a two-fold upregulation in KD-SDHB cells cultured in 3D (*p* < 0.009) (Appendix A).

To examine the role of glutamine, KD-SDHB and hPheo1 cells were cultured in a glutamine deficient medium. OCR showed a decreased activity in KD-SDHB cells (OCR mean = 105 ± 6.9 vs. 145 ± 8.9 in hPheo1 cells). All mitochondrial respiration parameters showed a significant decrease in OCR (*p* < 0.05) (Figure 6C). Besides, ECAR showed a decrease in glycolytic activity in KD-SDHB cells (ECAR mean = 21 ± 2 vs. 26 ± 2 in hPheo1 cells). In fact, glycolysis remained decreased (*p* < 0.001) in KD-SDHB. Interestingly, the non-glycolytic acidification decreased slightly (*p* = 0.056) (Figure 6D). We hypothesized that the mitochondrial respiration and the glycolytic pathway are dependent on glutamine to maintain energy in the deficient SDHB cells and that KD-SDHB cells enhance glutamine utilization compared to the control hPheo1 cells.

### 2.7. GLUD1 and SDHB Gene Expression in PCC/PGL

We investigated the correlation of GLUD1 mRNA expression with mRNA expression SDHB using available TCGA data. Expression of GLUD1 was higher in the tumors with low SDHB. There was a negative correlation between GLUD1 mRNA and SDHB mRNA expression (Pearson = −0.31, *p* = 0.03), confirming our findings. 

## 3. Discussion

In this study, we showed that SDHB deficient cancer cells exhibit a switch in the cellular metabolism from glycolysis to glutaminolysis to survive. Metabolic reprogramming has been proposed to be a hallmark of cancer and mitochondrial metabolic reprogramming exists in most tumors [18]. A defect in one of the mitochondrial enzyme complexes, i.e., SDHx (complex II), shows that the metabolic adaptation fuels cell growth and proliferation in PCCs/PGLs. In KD-SDHB cells, oxygen consumption is decreased, and succinate is accumulated, leading to HIF1a stabilization. The HIF1 complex transactivate target genes involved in metabolic rewiring that shifts cellular energy towards glycolysis [5]. Metabolic profiling of SDHB-mutated tumors shows a metabolic shift towards aerobic glycolysis as well as dependence on glutamine [6,19]. The high lactate level found in our KD-SDHB cells may result in the activation of at least two metabolic pathways. Lactate is generated from glucose during glycolysis and glutamate during glutaminolysis through the truncated TCA cycle. Based on cell proliferation and oxygen consumption, hPheo1 cells are not negatively affected by knocking down SDHB due to genetic haploinsufficiency. By examining whether the proliferation of KD-SDHB cells were dependent on glycolytic metabolism and mitochondrial respiration or not, we could observe that KD-SDHB, in those cells, seems to have higher oxygen consumption and ATP production, but glycolysis function is not as strongly activated as seen in hPheo1 cells. In contrast, we have shown that decreased expression in OXPHOS genes and increased expression in the glycolysis pathway may represent metabolic compensatory mechanisms that the tumor cells develop in response to KD-SDHB. We assume that the Warburg phenotype predominates in hPheo1, but the KD-SDHB cells prefer oxidative phosphorylation using glutamate. Similar observations have been suggested by Hujber et al. regarding human glioma cells containing IDH mutation [20]. Interestingly, Wang et al. showed that SDHx tumors present decreased complex II activity, associated with increased activity of the rest of OXPHOS complexes (I, III and IV), suggesting a compensatory effect [21]. The decrease in OXPHOS gene expression has been shown earlier in SDHx mutated PCCs/PGLs [22]. In contrast, increased gene expression has been observed in mutated SDHB PCCs/PGLs, associates with a decreased OXPHOS activity as a result of ROS accumulation [23].

In parallel, both hPheo1 and KD-SDHB cells seems to enhance glutamine utilization. We suggest that the flow to the TCA cycle increased in chromaffin cells with SDHB mutation, promoting mitochondrial oxidative phosphorylation, as suggested before for KD-SDHB PC12 cells (rat chromaffin cells) [6].

The OCR and ATP production are enhanced by glutamine, contributing to tumorigenesis [24]. Furthermore, the glutamine transporter level regulates the cellular ATP level induced by glutamine, implying that glutamine can be used as a source of energy via mitochondrial glutaminolysis [24,25].

We hypothesize that SDH-deficient cells adapt to a more rapid metabolism, which allows these cells to survive by shifting its metabolism to the use of an alternative fuel, glutamine.

Glutamine has an extensive role in cell metabolism, and disruption of the TCA cycle makes the cells more dependent on reductive carboxylation of glutamine instead of the oxidative metabolism of the TCA cycle [24].

Glutamine is an important amino acid that controls cell growth and metabolism via the activation of the mammalian/mechanistic target of rapamycin complex 1 (mTORC1), but the pathway remains not well defined. Glutamine is metabolized through glutaminolysis to produce α-ketoglutarate. In combination with leucine and by enhancing glutaminolysis and α-ketoglutarate production, glutamine activates mTORC1 [25,26].

mTORC1 is a regulator of several cellular processes involved in proliferation, cell growth and metabolism and controls cellular metabolism by regulating metabolic genes, such as peroxisome proliferator activated receptor γ coactivator-1 α (PGC-1α). It is reported earlier that breast cancer cells used PGC-1α to enhance oxidative phosphorylation, mitochondrial biogenesis, oxygen consumption rate, and promote metastasis [27]. 

On the other hand, it is known that glutamine is absorbed by tumor cells [6,28] and metabolized within the mitochondria via two deamination steps: (1) production of glutamate by glutaminase (GLS1/GLS2) through an irreversible reaction, (2) production of α-KG by glutamate dehydrogenase (GLUD1) [15]. The α-KG generated by glutaminolysis is used to fuel the TCA cycle and OXPHOS in tumor cell mitochondria.

The regulation of both mTORC1 and glutaminolysis suggests that they may regulate each other in promoting cell growth and cancer progression. mTORC1 also induces glutaminolysis by activating c-MYC, which is a transcriptional activator of SLC1A5, GLS, and GLUD1 transcription. In MYC-driven cells, glutamine uptake is enhanced where the glutamine transporter SLC1A5 expression is upregulated. Deamination of glutamine to glutamate in mitochondria is enhanced by c-MYC via the upregulation of GLS1. Conversion of glutamate into α-KG is mediated by GLUD1 with c-MYC-driven signaling activation. Through the promotion of glutamine metabolism, the c-MYC oncogene could partially modulate cancer metabolic reprogramming, favoring cancer cell growth and survival by the upregulation of the c-MYC-target and mTORC1 pathway in KD-SDHB hPheo1 cells, as is demonstrated from the microarray data, where GLUD1 expression is enhanced to induce glutaminolysis. We assume that an impaired TCA cycle lead the KD-SDHB cells to shift its metabolism from glycolysis to glutaminolysis. Thus, the mTORC1 pathway maybe activated to promote GLUD1 activity via a c-MYC-mediated increase of GLUD1 expression. The glutamine dehydrogenase activity is needed for tumor cell survival during glycolysis impairment and an altered expression of GLUD1 may affect cell proliferation, migration, and invasion [20,29,30]. 

Although in vitro cell cultures and tumor samples are clearly different, we have explored if high GLUD1 expression observed in hPheo1 cells are also expressed in human PCCs/PGLs (TCGA data). Expression of GLUD1 was higher in the low SDHB-expressed tumors. Our analysis suggests that enhanced GLUD1 expression is dependent on the downregulation of SDHB in PCC/PGL tumors, which thus may represent a potential biomarker and therapeutic target for SDHB-mutated tumors. Recently, the most reported GLUD1 inhibitor is EGCG (epigallocatechin gallate), targeting a group of enzymes that use NADPH as a cofactor [31], is under clinical trial for several tumors, i.e., prostate, breast, and colorectal cancers (https://clinicaltrials.gov/ct2/home) (accessed on 20 December 2021). In addition, two specific GLUD1 inhibitors (purpurin and a purpurin analogue, R162) are potent inhibitors in cancer cells [32].

Downregulation of SDHB activity in hPheo1 cells alter the adhesion to the extracellular matrix. Increased trypsin sensitivity and increased expression of matrisomal and cell adhesion genes suggest a malignant phenotype of the KD-SDHB cells, which has been observed earlier in breast cancer cells exhibiting more malignant phenotypes [33]. 

Cell migration or invasion in malignant PCCs/PGLs characterized by altered cellular adhesion and ECM remodeling have been observed, and may be attributed to SDHB mutations that lead to malignancy [34]. As for pseudohypoxic PCCs/PGLs with mitochondrial function, KO-SDHB showed improved invasion and malignant potential in murine chromaffin cells. [35]. 

The extracellular matrix (ECM) in solid tumors, similar to the other components of the tumor microenvironment (TME), differs considerably from those in normal organs. The ECM controls tumoral signals, transport mechanisms, metabolism, oxygenation, and immunogenicity. ECM exerts regulatory control over the tumor, influencing not only its malignancy and development, but also its responsiveness to therapy [36].

The interaction with ECM molecules may induce epithelial-to-mesenchymal transition (EMT) [37,38], which is a crucial cellular program that enables polarized epithelial cells to transition toward a mesenchymal phenotype with increased cellular motility [39]. EMT can be initiated by external signals, such as transforming growth factor (TGF)-β [40,41,42]. 

The mechanisms that govern EMT are being unraveled, and many effectors of EMT are known to modulate adhesion systems, remodel the actin cytoskeleton, and promote the mesenchymal phenotype [36,41]. Many solid tumors have elevated expression of ECM molecules, such as fibrillar collagens, fibronectin, elastin, and laminins [36,43,44]. As the major component of the tumor matrix, collagen greatly influences tumor invasion and prognosis. High collagen mRNA expression, as observed in our data (Figure 3A), abnormal types of fibrous collagen spindles, and enhanced expression of matrix proteases are all indicators of rapid collagen turnover in tumors [36,45]. Our results showed that the expression of matrix metalloproteinase (MMP) (collagenase) and lysyl oxidase (LOX) was elevated in KD-SDHB cells, which contributed to the hyperactive remodeling of collagen (Appendix A). In cancer stroma patients with upregulated expression of a combination of coding genes for collagen and collagenase, it has been shown that they exhibited poorer overall survival times [46]. 

Aberrant activation of the PI3K/Akt pathway is often observed during the EMT process and correlates with the progression of different types of human cancer. For example, in lung cancer, collagen I induces EMT by autocrine activation of TGF-β3 signaling and that PI3K is necessary for TGF-β up-regulation [41,47], similar to the observed activation collagen I and TGF-β3 observed in the KD-SDHB hPheo1 cells (Figure 3A). 

## 4. Materials and Methods

### 4.1. Cell Culture

The cell line was maintained as adherent monolayer cultures in RPMI 1640 medium (Gibco), supplemented with 5% heat inactivated FBS (Gibco), 1% L-glutamine (Gibco) and 1% penicillin-streptomycin (Gibco). Cells were seeded at 2 × 10^5^/mL in T75 tissue culture flasks and the media was changed once a week. 

### 4.2. SDHB Knockdown with CRISPRcas9

The hPheo1 cell line was transfected with the True cut cas9 protein v2 and True guide synthetic gRNA (Thermofisher) and analyzed at 48 to 72 h post-transfection. The genomic cleavage efficiency was measured by the GeneArt^®^ Genomic Cleavage Detection kit (Thermofisher), based on the relative gel band intensity, using the microfluidic capillary electrophoresis instrument, Qiaxcel (Qiagen). Transfected cells were collected as deficient *SDHB*-hPheo1 polyclonal cells. 

### 4.3. DNA Extraction and Sequencing Confirmation 

Genomic DNA was extracted from culture cells using the GeneArt^®^ Genomic Cleavage Detection kit (Thermofisher) following the manufacturer’s instruction and quantified using NanoDrop1000 (Thermo Scientific). PCR amplification using specific primers for the SDHB gene (F: GCCTCCTGGGTTCAAGCAAT and R: TGTTGGATATTGAATGCCTGCC) was used and direct sequencing of PCR products was performed with the ABI Prism BigDye Terminator Cycle Sequencing Ready Reaction Kit (ABI PRISM/Biosystems) and the products were resolved on ABI PRISM 3500 (Applied Biosystems). The Blast homology searches were performed using the program available at the National Center for Biotechnology Information web site in comparison with the updated sequence (GenBank Accession No. NM_003000, build GRCh37 (hg19).

### 4.4. Spheroid Formation

Tumor spheroids were generated by seeding 10,000 cells/well of hPheo1 cells in ultra-low attachment (ULA) 96-well round-bottomed plates (Corning, Amsterdam, The Netherlands), centrifuged for 5 min/900 g and incubated for 3 to 6 days to allow formation into a solid spheroid. Images were captured with a Leica DMI 4000 inverted fluorescence microscope (Leica Microsystems, Wetzlar, Germany).

### 4.5. RNA Extraction 

Trizol was used to lyse the cells (Invitrogen) and the RNeasy minikit (Qiagen, Hilden, Germany) was used for RNA purification and quantified with a NanoDrop1000 instrument. RNA integrity was checked with the RNA 6000 Nano kit using the Bioanalyzer 2100 instrument (Agilent, Böblingen, Germany). For all samples involved in the microarray analysis, the RIN value was ≥8.5. 

### 4.6. SDHB Gene Expression

cDNA was generated using the Maxima First Strand cDNA Synthesis Kit (Thermo Scientific). Complementary DNA (10 ng per sample) served as a template for quantitative PCR with the use of a QX200 digital droplet PCR (ddPCR) system and specific primers for SDHB genes (assay ID: Hs01042481_m1) labelled with FAM dye and GUSB (β-Glucuronidase) (assay ID: dHsaCPE5050189) labeled with HEX and used as a reference gene. The gene expression data were processed using Quanta Soft software (Bio-Rad). Each reaction was performed in triplicate.

### 4.7. Western Blotting

Proteins were extracted using standard protocol for RIPA buffer. The resulting pellet was resuspended in a 50 mM Tris buffer (pH 7.4) with protease inhibitor and protein concentration and is determined using BCA Protein assay kit (ThermoFisher).

After denaturation for 5 min at 95 °C, sodium dodecyl sulphate (SDS)–polyacrylamide gel electrophoresis (4–15% Mini-PROTEAN^®^ TGX™ Precast Protein Gels, #4561083, BioRad) was performed using equal amounts of proteins (25 ng), followed by transfer to the PVDF membrane (Trans-Blot^®^ Turbo™ Mini PVDF Transfer Packs #1704156, BioRad). After blocking the reaction with 5% milk, the membrane was incubated with the primary antibodies: anti-SDHB diluted titer of 1:1000 (ab14714, Abcam) and anti-alpha-Tubulin diluted titer of 1:7000 (DM1A, Invitrogen). The PVDF membrane was further incubated with a mouse secondary antibody (1:2000) (bs-029G-HRP, Bioss) and developed using an enhanced chemiluminescence system (Clarity Western ECL Substrate, Bio-Rad) according to the manufacturer’s instructions. To analyze the bands, we scanned the blots and quantified the bands using densitometry (ChemiDoc, Bio-Rad). To compare protein expression among the different samples, the average intensity of the immunoblot signal was multiplied by the number of pixels in that area and normalized to alpha-Tubulin expression.

### 4.8. Biochemical Analysis

hPheo1 control and KD-SDHB cells were plated at a density of 3×10^5^ on 6 well plates (corning) and cultured for 72 h post seeding. For metabolic analysis of extracellular metabolites, culture media were collected, incubated with one volume of 100% cold methanol for 30 min at −20 °C, and centrifuged at 16,000× *g* at 4 °C for 20 min. The supernatant was collected and stored at −80 °C until further analysis. 

Pyruvate and lactate levels were determined in the samples with ISCUSflex Analyzer (CMA Microdialysis AB, Solna, Sweden). The detection intervals were as follows: 0.1–12 mmol/L for lactate and 10–1500 µmol/L for pyruvate.

### 4.9. Microarray Analysis

RNA, from KD-SDHB and control hPheo1 cells, was reverse-transcribed with the One-Cycle cDNA Synthesis kit. cDNA was hybridized to a HuGene_1_0_st_v1 microarray chip (Affymetrix). Arrays were washed, stained, and scanned according to standard protocols supplied by the manufacturer. After scanning, files were normalized and summarized to probe set expression values using the robust microarray average (RMA) [48]. Data preprocessing and calculation of log2 fold-changes between samples expression values were performed in Transcriptome Analysis Console (TAC) Software (Affymetrix). Transcripts with two- or greater-fold change in expression as well as the student *t*-test output of *p* < 0.05 were used to identify significant differential expression. This comparison identified genes which were overexpressed or underexpressed, as well as those which were either present or absent in silenced cells. Gene Set Enrichment Analysis (GSEA) was computed referring to the Molecular Signatures database (MSigDB) for the “Hallmark” and “C2: Canonical pathways” gene sets. 

GSEA performed 1000 permutations. The maximum and minimum sizes for gene sets were 500 and 15, respectively. Cutoff for significant gene sets was a false discovery rate < 1%.

### 4.10. RT-qPCR

The 2^−ΔΔCT^ method was used for real-time reverse transcriptase quantitative PCR (RT-qPCR) for *CCND1*, *ACTA2*, *COL1A2*, *COL6A4*, *TGFB3*, *PCG1-α*, *GLUT1*, *HK2*, *GPI*, *ALDOA*, *ALDOC*, *PKM2*, *LDHA*, *SLC1A5*, *GLS1*, *GLUD1*, *GLUL*, and *GOT2* mRNA expression. Primers were designed to span an exon–exon junction (see Table 1 for primer sequences). β-glucuronidase (*GUSB*), hypoxanthine phosphoribosyl transferase 1 (*HPRT*), and β-actin (*ACTB*) were used as reference genes. On a 7500 Fast Real-Time PCR System, RT-qPCR reactions were run using Power SYBR^®^ Green PCR Master Mix (Applied Biosystems). All samples were performed in technical triplicates. 

### 4.11. Cell Proliferation 

A total of 3000 cells/well were plated on 96 well plates. From day 1 to day 3, cell proliferation was assessed with Cell Titer-Glo Luminescent Cell Viability Assay (Promega) following the manufacturer´s protocol. 

### 4.12. Clonogenic Assay 

The clonogenic assay was carried out as previously described [49]. A total of 100 cells/well were seeded and incubated at 37 °C with 5% CO_2_ for 14 days. Then, the cultures were stained with 0.5% crystal violet for 30 min and colonies were counted using ImageJ software 1.50i. Three independent experiments were performed with three technical replicates in each experiment.

### 4.13. Differential Trypsinization

hPheo1 cells and KD-SDHB hPheo1 cells were grown to confluency in a 9 cm^2^ dish, washed with PBS and treated with 0.05% trypsin and 0.18 mM EDTA (Gibco) diluted in PBS at 37 °C. 

The “trypsin-sensitive cells” was generated by transferring cells detached by 1 min trypsinization to a new dish. The “trypsin-resistant cells” were generated by continuing the culturing of the starting population for 30 min, washing with PBS and discarding the detached cells [33,50].

### 4.14. Mitochondrial Content Measurement

Mitochondrial DNA copy number analysis was performed by ddPCR as described previously by Memon et al. [51], using probes targeting the mitochondrially encoded NADH dehydrogenase 1 (MT-ND1) (assay ID: dHsaCPE5029120, Bio-Rad) gene labelled with FAM fluorophore (mitochondrial DNA) and the eukaryotic translation initiation factor 2C, 1 (EIF2C1) (assay ID: dHsaCP1000002, Bio-Rad) and ribonuclease P/MRP 30 kDa subunit (RPP30) (assay ID: dHsaCP1000485, Bio-Rad) genes (nuclear DNA) labelled with HEX.

### 4.15. Mitochondrial Membrane Potential (ψm)

Mitochondrial membrane potential (ψm) was measured using the tetramethyl rhodamine ethyl ester (TMRE; Abcam ab113852, Cambridge, United Kingdom) reagent. Cells were grown in a 96-well, clear bottom, black plate. The media was gently aspirated from the correspondent wells and replaced by 100 μLof 20μM FCCP and incubated at 37 °C for 10 min and washed 3 times. Thereafter, 100 μLof 60 nM TMRE dissolved in cell culture media was added to all wells, and incubated at 37 °C for 30 min, protected from light. Cells were gently washed 3 times and fluorescence was measured on a Spark10 plate reader with excitation 530 nm and emission 580 nm.

### 4.16. Measurements of Glycolytic and OXPHOS Activities

The Seahorse Extracellular analyzer XF24 (Agilent Technologies, Wilmington, DE, United States) was used. A total of 90,000 cells/well in 250 μL culture medium were plated, with the XF24 plate containing 4 blank controls, 24 h before the measurement started. 

To test cellular metabolic activities, 1 h before the measurements, medium was replaced with 500 µLof Seahorse assay media (1 mM pyruvate, 2 mM glutamine, 10 mM glucose, pH 7.4) and incubated at 37 °C without CO_2_. The cartridges were loaded with Oligomycin (1 uM) (complex V inhibitor), FCCP (1 uM) (respiratory uncoupler), and rotenone/antimycin A (0.5 uM) (complex I/III inhibitors) from the Seahorse XF Cell Mito Stress Test Kit (Agilent). Oxygen consumption rate (OCR) and ATP production values were measured by the XF24 Analyzer. The addition of FCCP accelerates oxygen consumption to a maximum level, whereas complex I/III inhibitors completely abolish mitochondrial respiration. Thus, the difference in OCR between FCCP- and rotenone/antimycin-treated states indicates the maximum OXPHOS capacity. 

To test for glycolytic activities, cells were starved in glucose-free medium for 1 h (RPMI with 2 mM glutamine), after which glucose was reintroduced (10 mM). Cells were then treated with 1 uM Oligomycin (complex V inhibitor) and with 50 mM 2-DG (2-Deoxy- d-glucose, glycolytic inhibitor) from the Seahorse XF Glycolysis Stress Test Kit (Agilent). Thus, the Agilent Seahorse XF24 Analyzer measured the extracellular acidification rate (ECAR).

To test the role of glutamine, the cells were cultured in glutamine-deprived medium for 1 h, after which mitochondrial respiration and glycolytic activity were measured. 

### 4.17. PCC/PGL TCGA Cohort Analysis

The gene expression dataset from the previously published RNA-sequencing Pheochromocytomas and Paragangliomas was downloaded from The Cancer Genome Atlas (TCGA) project [52]. Data on 178 samples were normalized and analyzed. The correlation analysis between gene expression values was assessed by the non-parametric Spearman correlation test.

### 4.18. Statistical Analysis

A students’ *t*-test was used to compare the means of the two groups. ANOVA was used to compare multiple groups with a two-tailed student’s *t*-test and Bonferroni correction. We consider p values greater than 0.05 to be statistically significant. Graphs were generated using GraphPad Prism software V5.0. Three independent tests were conducted on each experiment.

## 5. Conclusions

In normal aerobic respiration, SDH is an important mitochondrial enzyme involved in the Krebs cycle and the electron transport chain. In addition, SDH has tumor-suppressive effects. SDHB inactivation results in the accumulation of succinate and induces the stabilization of hypoxia-inducible factor (HIF) via competitive inhibition of HIF prolyl-hydroxylases [53,54]. We believe that stabilized HIF activates pseudo-hypoxic signaling and leads the dysregulation of cellular proliferation, adhesion, and glucose metabolism via the PI3K/Akt/mTORC1 pathway. A limitation of this study is that many analyses were only performed on the mRNA level. Future studies would need to confirm the results even on the protein level. Besides, we assume that the glutaminolysis pathway contributes to SDHB-mutant chromaffin cell malignancy. We believe that it provides an opportunity to properly investigate for targeted therapy, particularly utilizing glutaminolysis inhibitors.

## Figures and Tables

**Figure 1 ijms-23-00560-f001:**
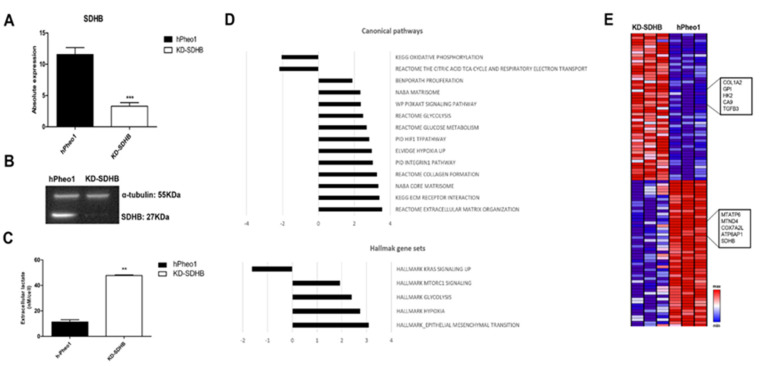
Effects of knockdown SDHB on hPheo1 cell line. (**A**) mRNA expression in control (black bar) and KD-SDHB (white bar) hPheo1 cells is presented as absolute quantification with ddPCR; each bar represents the mean ± standard deviation of three independent experiments, *** (*p* < 0.0001), student’s *t*-test. (**B**) Protein expression in hPheo1 (control) and KD-SDHB analyzed with Western blot using antibodies against SDHB and Alpha-tubulin. (**C**) Lactate concentrations in the medium were measured in KD-SDHB cells and hPheo1 control cells. *p* value calculated with student’s *t*-test indicated as ** (*p* < 0.001). (**D**) Top-ranked gene sets in the Canonical Pathways and Hallmark sets (up/down regulated). (**E**) Hierarchical clustering of hPheo1 and KD-SDHB cell lines based in mRNA expression levels. The normalized expression 2-log is displayed with z scores and altered expressed genes belong to glycolysis, hypoxia, cell proliferation, cell differentiation, and oxidative phosphorylation gene pathways.

**Figure 2 ijms-23-00560-f002:**
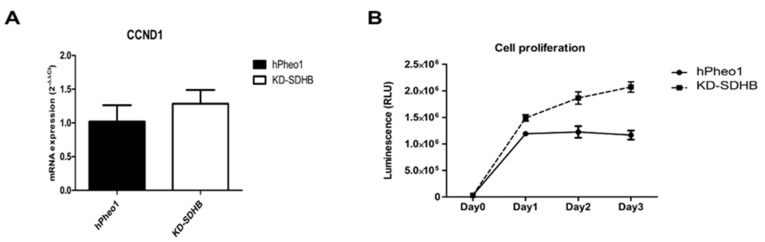
(**A**) Expression of CCND1 mRNA using RT-qPCR. (**B**) Cell growth test at 1, 2, and 3 days after plating of hPheo1 and KD-SDHB hPheo1 cells, based on luminescence generated by living and proliferating cells. Data represent the mean ± standard deviation (*n* = 3), student’s *t*-test.

**Figure 3 ijms-23-00560-f003:**
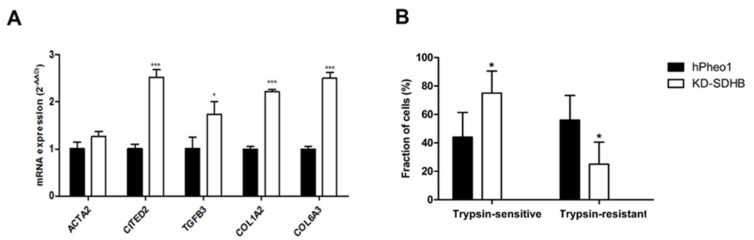
(**A**) Cell adhesion related genes upregulated by KD-SDHB. Relative mRNA expression of *ACTA2*, *CITED2*, *TGFB3*, *COL1A2*, and *COL6A3* in KD-SDHB hPheo1 cells by RT-qPCR analysis. (**B**) Cell counts for trypsin-sensitive and trypsin-resistant hPheo1 cells presented as percentages of total cell number. Each bar represents the mean ± standard deviation (*n* = 3), * (*p* < 0.05) and *** (*p* < 0.0001), student’s *t*-test.

**Figure 4 ijms-23-00560-f004:**
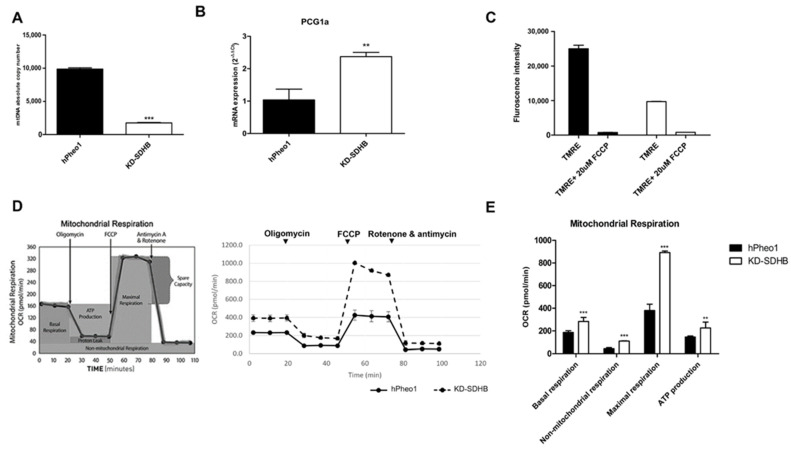
(**A**) Absolute mtDNA copy number in KD-SDHB and control cells (hPheo1 cell line). *p*-value was calculated by student’s *t*-test, *** (*p* < 0.0001). (**B****)** Expression of PCG1α mRNA using RT-qPCR (mean ± standard deviation) p-value was calculated by student’s *t*-test, ** (*p* < 0.001). (**C**) Chart showing mean fluorescent intensity ± standard deviation from triplicate measurements of TMRE stained hPheo1cells +/− treatment with FCCP. TMRE assay was performed to measure mitochondrial membrane potential in hPheo1 cell line (KD-SDHB and control). (**D**) The Agilent Seahorse XF24 analysis of mitochondrial function shows acute disruption of ATP-linked oxygen consumption in hPheo1 wild-type and KD-SDHB cells. Seahorse XF24 analysis using the Cell Mito Stress Test kit was performed (normalized to cell number). Respiratory chain stressor compounds were added to the culture at the indicated time points. (**E**) Individual parameters of mitochondrial function: basal respiration, nonmitochondrial oxygen consumption, maximal respiration, and ATP production, were calculated according to manufacturer’s protocol. Values are means ± standard deviation of at least three technical replicates at each time point (see experimental procedures for details). Student’s *t*-test *p* values indicated as ** (*p* < 0.001) and *** (*p* < 0.0001).

**Figure 5 ijms-23-00560-f005:**
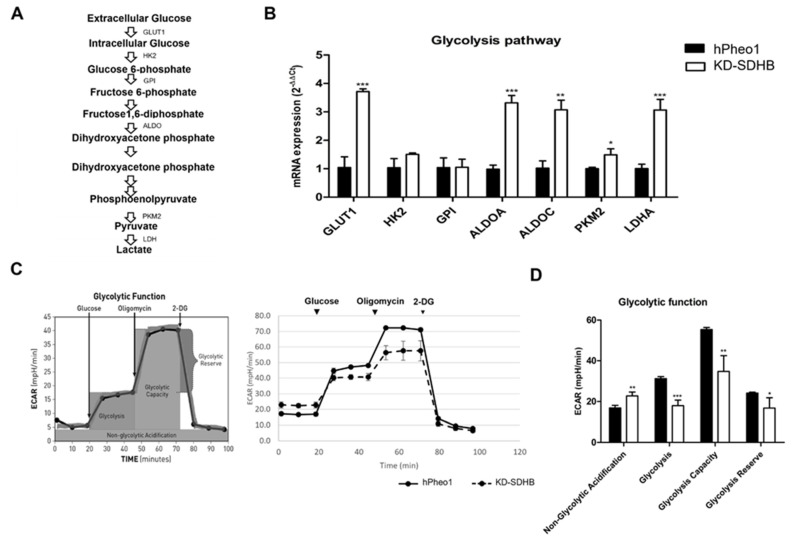
(**A**) Summary indicating the glycolysis pathway. (**B**) Upregulation of the glycolysis gene set by knocking down SDHB. Each bar represents the mean ± standard deviation, * (*p* <0.01), ** (*p* < 0.001), *** (*p* < 0.0001), student’s *t*-test. (**C**) The Agilent Seahorse XF24 analysis of glycolytic function shows an alteration in the ECAR in KD-SDHB cells compared to hPheo1 cells. Seahorse XF24 analysis performed using Glycolysis-stress test kit (normalized to cell number). Glycolysis stressor compound was added in specific point time. (**D**) Specific parameters for non-glycolytic acidification, glycolysis, glycolytic capacity, and glycolytic reserve were calculated according to manufacturer’s protocol. Values are means ± standard deviation of at least three technical replicates at each time point (see experimental procedures for details). Student’s *t*-test *p* values indicated as * (*p* <0.01), ** (*p* < 0.001) and *** (*p* < 0.0001).

**Figure 6 ijms-23-00560-f006:**
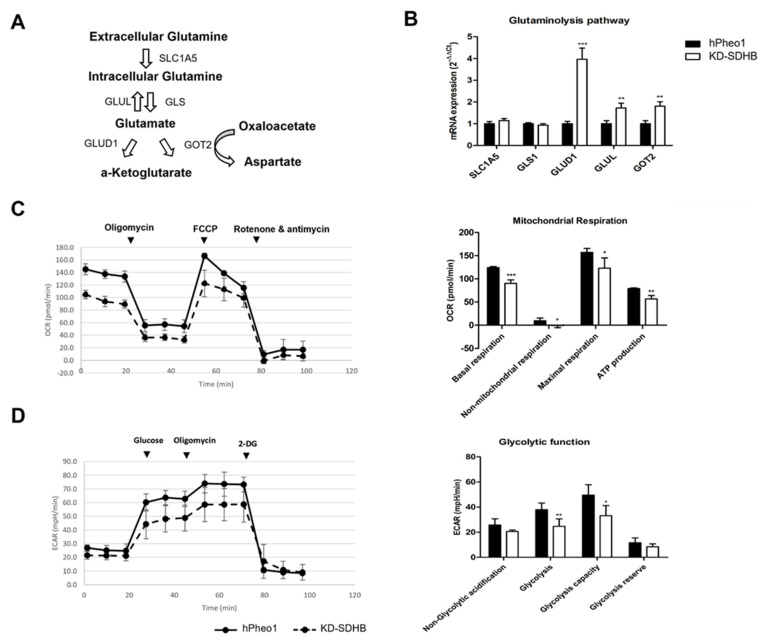
(**A**) Summary indicating the glutaminolysis pathway. (**B**) Upregulation of the gene set of glutaminolysis by knocking down SDHB. Each bar represents the mean ± standard deviation, ** (*p* < 0.001) and *** (*p* < 0.0001), student’s *t*-test. (**C**) The Agilent Seahorse XF24 analysis of mitochondrial function shows an alteration in OCR in KD-SDHB cells compared to acute disruption of ATP-linked oxygen hPheo1 cells cultured in deprived media of glutamine. Basal respiration, non-mitochondrial oxygen consumption, maximal respiration, and ATP production were calculated according to manufacturer’s protocol. Student’s *t*-test *p* values indicated as * (*p* < 0.01), ** (*p* < 0.001) and *** (*p* < 0.0001). (**D**) The Agilent Seahorse XF24 analysis of glycolytic function shows an alteration in the ECAR in KD-SDHB cells compared to hPheo1 cells cultured in deprived media of glutamine. Specific parameters for non-glycolytic acidification, glycolysis, glycolytic capacity, and glycolytic reserve were calculated according to manufacturer’s protocol. Student’s *t*-test *p* values indicated as * (*p* < 0.01) and ** (*p* < 0.001).

**Table 1 ijms-23-00560-t001:** List of primers used for RT-qPCR; primer sequences and amplicon lengths, with *GUSB (G)*, *HPRT1 (H)*, or *ACTB (A)* as reference genes, are provided.

Gene	Sequence (5′→3′)	Amplicon Length (bp)
*CCND1*	F: GAGGAGCTGCTGCAAATGG R: CGGCCAGGTTCCACTTGA	58 *(G)*
*ACTA2*	F: CGGGAGAAAATGACTCAAATTATGTT R: CATACATGGCTGGGACATTGAA	62 *(G)*
*PCG1-α*	F: TGAGAGGGCCAAGCAAAG R: ATAAATCACACGGCGCTCTT	*64 (G)*
*TGFB3*	F: TATGTGATTGCCATCTTTGCC R: TGGACCTGCAGTGGCTAAACA	54 *(G)*
*COL1A2*	F: ACCACAGGGTGTTCAAGGTG R: CAGGACCAGGGAGACCAAAC	149 *(G)*
*COL6A3*	F: CCTAACCACATATGTTAGTGGAGGT R: GAATGTCTCGCTTGCTCTCTG	71 *(G)*
*GLUT1*	F: TTGTGGGCATGTGCTTCCAG R: ATCGAAGGTCCGGCCTTTAG	134 *(G)*
*HK2*	F: GGTGGACAGGATACGAGAAAACR: ACATCACATTTCGGAGCCAG	141 *(G)*
*GPI*	F: AAATCGCCCAACCAACTC R: ATGATGCCCTGAACGAAG	102 *(G)*
*ALDOA*	F: TTGTGGGCATCAAGGTAG R: TAGTCTCGCCATTTGTCC	60 *(G)*
*ALDOC*	F: ATCGTCGTGGGCATCAAGG R: TTGGGCACAGCGTTCTGAG	105 *(G)*
*PKM2*	F: ATCGTCCTCACCAAGTCTGG R: GAAGATGCCACGGTACAGGT	126 *(A)*
*LDHA*	F: GAG GTG ATC AAA CTC AAA GGC T R: CAT GGT GGA AAC TGG GTG C	111 *(G)*
*SLC1A5*	F: CTCGATTCGTTCCTGGATCTT R: GTTCCGGTGATATTCCTCTCTTC	107 *(A)*
GLS1	F: TCTACAGGATTGCGAACGTCT R: CTTTGTCTAGCATGACACCATCT	100 *(A)*
GLUD1	F: GGTCATCGAAGGCTACCG R: TCAGTGCTGTAACGGATACCTC	75 *(A)*
GLUL	F: CCTGCTTGTATGCTGGAGTC R: GATCTCCCATGCTGATTCC	105 *(A)*
GOT2	F: GACCAAATTGGCATGTTCTGT R: CGGCCATCTTTTGTCATGTA	95 *(A)*
ACTB	F: CTCTTCCAGCCTTCCTTCCT R: AGCACTGTGTTGGCGTACAG	116
*HPRT1*	F: ATGGACTAATTATGGACAGGACTGAA R: CTCCCATCTCCTTCATCACATCT	60
*GUSB*	F: CAAGACAGTGGGCTGGTGAATTA R: CTTGAACAGGTTACTGCCCTTGAC	55

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
