# Peer review of "Loss of SDHB Induces a Metabolic Switch in the hPheo1 Cell Line toward Enhanced OXPHOS"

_ijms, 2022, doi:10.3390/ijms23010560_

Round 1
Reviewer 1 Report
In the manuscript entitled “Loss of SDHB induces a metabolic switch in hPheo1 cell line toward enhanced OXPHOS” the authors examine complex consequences of SDHB deficiency in human pheochromocytoma cell line. The research is well-established, addresses current questions about PPGL tumorigenesis and seeks answers with various research methods and proves that SDH associated PCC behaves more aggressively in vitro as well compared to „sporadic” PCC.
I have a few concerns:
- The citations in text vary: in the first paragraph authors and dates are shown whereas later on only numbers in brackets.
- Typos should be corrected. For example:
-Figure 1 legend (page 3): *** (p < 0.0001) „p” should be in italics.
-Further in the text (page 3, 4, 5, 8) „p” should be in italics
-page 3 first paragraph: „up regulated” should be merged
-page 4 first paragraph: „in vitro” should be in italics
-page 4: chapter 2.2 first sentence: dot is missing at the end of the sentence.
-page 5: Pyruvate kinase: P should not me in italics.
-the author uses both „tumor” and „tumour” in the text. One should be selected.
Questions:
- „By examining whether the proliferation of KD-SDHB cells were dependent on glycolytic metabolism and mitochondrial respiration or not, we could observe that KD-SDHB in those cells seems to have higher oxygen consumption and ATP production, but glycolysis function is not as strongly activated as seen in hPheo1 cells.” This statement can be misunderstood as it is in contrast with the results of the extracellular metabolite measurements, gene expression analysis and ECAR measurements, but it is in line with the hexokinase-2 inhibition experiments. Therefore this section should be explained in more detail.
- Based on the OCR measurements the authors concluded that the mitochondrial OXPHOS activity is increased in KD-SDHB cells whereas the gene expression analysis shows the opposite. The authors’ opinion about the reason for this should be considered to be implemented in the discussion and confronted with other previous publications’ results (e.g. PMID: 22859959).
- „The enhanced expression of genes involved in glycolysis pathway was accompanied with an increase in the glycolytic activity (ECAR) in KD-SDHB cells (ECAR mean= 23±1.9 vs 17.3±1.2 in hPheo1 cells).” whereas on figure 5. panel D. an increase in the non-glycolytic acidification can be seen compared to hPheo cells and the opposite can be seen regarding glycolysis, glycolysis capacity and reserve. This should be clarified.
- “Expression of GLUD1 was higher in the SDHB low expressed tumors. There was a positive correlation between GLUD1 mRNA and SDHB mRNA expression (Pearson= 0.31, p= 0.03).” Based on the results a negative correlation would be expected. What is the authors’ opinion on this?
- In the discussion the authors may want to comment on the relation between the impaired TCA cycle, tumorigenesis and glutamate dehydrogenase function, as GLUD expression has been previously linked to tumor progression (e.g. IDH mutated glioblastomas). A few examples: IDH deficiency+glutamine/glutamate metabolism PMID: 30404651. GLD1+cancer: PMID: 25670081, PMID: 23663782, PMID: 25947346. IDH mutation + GLUD2 inhibition: PMID: 25225364.
- If the authors aware of any ongoing pre-clinical or clinical study that targets glutamate dehydrogenase as an antitumor therapy it should be considered to be inserted in the discussion.
Reviewer 2 Report
Comments:
- Does SDHB antibody from Abcam recognize aa 54 protein? If yes, aa 54 band should be added to the Figure 1B.
- Does aa 54 protein localize in mitochondria? please show images.
- Any western blot data for Figures 3A, 5B, and 6B?
Round 2
Reviewer 2 Report
Comments:
Since authors cannot confirm whether Abcam antibody recognize 54 aa and lack of western blot analysis data, I suggest authors discuss these 2 issues in discussion or results section as the weakness of the current study.
